# Riboflavin (Vitamin B2) Deficiency Induces Apoptosis Mediated by Endoplasmic Reticulum Stress and the CHOP Pathway in HepG2 Cells

**DOI:** 10.3390/nu14163356

**Published:** 2022-08-16

**Authors:** Bo Zhang, Jun-Ting Cao, Yong-Bao Wu, Ke-Xin Gao, Ming Xie, Zheng-Kui Zhou, Jing Tang, Shui-Sheng Hou

**Affiliations:** 1State Key Laboratory of Animal Nutrition, Key Laboratory of Animal (Poultry) Genetics Breeding and Reproduction, Ministry of Agriculture and Rural Affairs, Institute of Animal Sciences, Chinese Academy of Agricultural Sciences, Beijing 100193, China; 2Key Laboratory of Feed Biotechnology of Ministry of Agriculture and Rural Affairs, Feed Research Institute, Chinese Academy of Agricultural Sciences, Beijing 100081, China

**Keywords:** riboflavin deficiency, apoptosis, endoplasmic reticulum stress, CHOP pathway

## Abstract

Riboflavin is an essential micronutrient and a precursor of flavin mononucleotide and flavin adenine dinucleotide for maintaining cell homeostasis. Riboflavin deficiency (RD) induces cell apoptosis. Endoplasmic reticulum (ER) stress is considered to induce apoptosis, and C/EBP homologous protein (CHOP) is a key pathway involved in this process. However, whether RD-induced apoptosis is mediated by ER stress and the CHOP pathway remains unclear and needs further investigation. Therefore, the current study presents the effect of RD on ER stress and apoptosis in the human hepatoma cell line (HepG2). Firstly, cells were cultured in a RD medium (4.55 nM riboflavin) and a control (CON) medium (1005 nM riboflavin). We conducted an observation of cell microstructure characterization and determining apoptosis. Subsequently, 4-phenyl butyric acid (4-PBA), an ER stress inhibitor, was used in HepG2 cells to investigate the role of ER stress in RD-induced apoptosis. Finally, *CHOP* *siRNA* was transfected into HepG2 cells to validate whether RD triggered ER stress-mediated apoptosis by the CHOP pathway. The results show that RD inhibited cell proliferation and caused ER stress, as well as increased the expression of ER stress markers (CHOP, 78 kDa glucose-regulated protein, activating transcription factor 6) (*p* < 0.05). Furthermore, RD increased the cell apoptosis rate, enhanced the expression of proapoptotic markers (B-cell lymphoma 2-associated X, Caspase 3), and decreased the expression of the antiapoptotic marker (B-cell lymphoma 2) (*p* < 0.05). The 4-PBA treatment and *CHOP* knockdown markedly alleviated RD-induced cell apoptosis. These results demonstrate that RD induces cell apoptosis by triggering ER stress and the CHOP pathway.

## 1. Introduction

Riboflavin is a precursor of flavin mononucleotide (FMN) and flavin adenine dinucleotide (FAD), which act as flavoprotein co-factors. Riboflavin is involved in a wide range of biological processes, such as the antioxidant system, protein folding, oxidative phosphorylation, fatty acid beta oxidation, and the tricarboxylic acid cycle [1,2,3,4,5,6,7]. Riboflavin deficiency (RD) is prevalent globally, particularly in regions with low intakes of dairy products and meats [8,9]. Additionally, the deficiency of this vitamin causes numerous health problems, such as growth depression, anemia, gastrointestinal tract disorders, and stomatitis in humans and animals, due to the impaired aforementioned biological processes [4,10,11,12,13,14]. In vivo, RD also inhibits cell proliferation and triggers apoptosis [15,16,17,18,19,20].

Riboflavin is involved in newly synthesized protein folds in the ER [7], as ER oxidoreductase 1 (Ero1) is a FAD-dependent enzyme that catalyzes the disulfide bond formation of newly synthesized proteins. Furthermore, riboflavin plays an important role in maintaining ER redox balance because of the glutathione reductase (GR) flavoprotein, which scavenges active oxygen species (ROS) from electrons transferred in disulfide bond formation [7,21,22,23,24]. The inhibition of disulfide bond formation and the accumulation of ROS block protein folding in the ER, which then triggers ER stress [25,26]. Theoretically, RD impairs protein folding due to reduced flavoproteins and redox imbalance, triggering an ER stress response. Indeed, previous studies have determined that RD markedly diminishes GR activity and glutathione (GSH) content [3,27], and downregulates the expressions of Ero1 and PDI [10,19,28]. Previous studies showed that RD could impair the folding and secretion of apolipoprotein-B100 (ApoB-100) and Interleukin-2 (IL-2) in human and mouse cells [15,17,29]. Additionally, it has been shown that RD can trigger ER stress [15,17].

Several studies indicate that RD can induce apoptosis [18,19,20], but the precise molecular mechanisms involved are not completely understood. It is well established that persistent or severe ER stress can induce apoptosis [30,31]. Previous studies have reported that RD hampers protein folding and triggers ER stress [15,17], which may lead to apoptosis. Thus, it was hypothesized that RD-induced apoptosis is mediated by ER stress. This remains to be confirmed. The C/EBP homologous protein (CHOP) pathway is one of the signaling pathways triggered by ER stress. It initiates a cascade of B-cell lymphoma 2-associated X (BAX) activation, B-cell lymphoma 2 (BCL2) inhibition, and caspase 3 (CASP3) activation, thus provoking cell apoptosis [32,33,34,35,36]. It has been shown that riboflavin depletion causes ER stress and apoptosis, as well as the up-regulation of CHOP [15,16,17,18,19]. Therefore, it was hypothesized that RD-induced apoptosis is mediated by the CHOP pathway, which remains to be confirmed.

In this study, we hypothesize that RD induces apoptosis mediated by ER stress and the CHOP pathway. To confirm the hypothesis, a riboflavin-deficient HepG2 cell model was constructed to investigate the effects of RD on ER stress and cell apoptosis. Then, a HepG2 cell model of ER stress alleviation was generated by applying 4-phenyl butyric acid (4-PBA) to explore the effects of ER stress inhibition on RD-induced cell apoptosis. Finally, a cell model of low CHOP expression was produced by siRNA transfection to explore the effect of CHOP interference on RD-induced apoptosis. Thus, this study systemically analyzes the role of ER stress and the CHOP pathway in RD-induced apoptosis.

## 2. Materials and Methods

### 2.1. Cell Culture and Experimental Protocols

HepG2 cells were purchased from the American Type Culture Collection (Manassas, VA, USA). The riboflavin-free medium was customized from Thermo Fisher Scientific (Waltham, MA, USA). The RD group medium consisted of the Rf medium + 5% fetal bovine serum (FBS, Gibco) + 4 mM L-glutamine (Solarbio, Beijing, China) + 1 mM sodium pyruvate (Solarbio) + 1% penicillin /streptomycin (Procell, Wuhan, China). The control (CON) medium was consistent with the RD medium; however, it was supplemented with 1000 nM riboflavin (Sigma Aldrich, St. Louis, MO, USA). The FBS contained 91.0 nM riboflavin determined by a high-performance liquid chromatography procedure, according to the method previously described [11,37]. The riboflavin concentration in the RD and CON mediums containing 4.55 and 1005 nM were formulated. Then, HepG2 cells were seeded at a density of 1 × 10^5^ cells /mL in 6-well plates and randomly assigned to 2 groups (n = 6 /per group), which were cultured in the 2 mediums mentioned above: the RD and CON groups. Following the completion of the incubation period, the cells were collected for subsequent ER stress and apoptosis analyses, and this was a preliminary study. Subsequently, to explore whether RD-induced apoptosis was mediated by ER stress, we constructed a cell model of the inhibition of ER stress. 4-phenyl butyric acid (4-PBA), an ER stress inhibitor, was purchased from Solarbio. It was added into the Rf medium at dosages of 0, 0.5, 2.5, and 5 mM. The cells were cultured in the above medium for 4 d for the ER stress-suppression assay and were subjected to subsequent apoptosis analyses. Finally, to explore whether ER stress-regulated apoptosis induced by RD was via targeted CHOP, we constructed a cell model of *CHOP* knockdown. Small, interfering RNA (*siRNA*) targeting human *CHOP* and a siRNA-negative control (NC) were designed and synthesized by Sangon Biotech (Shanghai, China). RNA-silencing sequences are listed in Table 1. The cells were passaged and cultured for 24 h, reaching a cell density of 40~50%. Next, the cells were transfected with *siRNA* using the Lipo8000TM transfection reagent (Beyotime Biotechnology, Shanghai, China), following the manufacturer’s instructions. The HepG2 cells were cultivated for an additional 3 d and then used for subsequent experiments.

### 2.2. Cell Structure and Microstructure Observations

The cultured 4 d cellular microstructure was examined by transmission electron microscopy (TEM, Hitachi, Tokyo, Japan), for the preparation of the electron microscopy sample. The cells were then washed with 0.1 M phosphate buffer 3 times, fixed in 1% osmic acid at 4 °C for 2 h, and embedded in agarose. Subsequently, the cells were dehydrated with alcohol gradient, embedded in Epon-Araldite 812 (Spi-Chem, West Chester, PA, USA) resin for penetration, placed in a model for polymerization, sectioned with an ultramicrotome (Leica, Vienna, Austria), and counterstained with 3% uranyl acetate and 2.7% lead citrate. Finally, these cell sections were observed under a TEM.

### 2.3. Cell Number and Viability

The cells were detached from 6-well plates by using trypsin (Solarbio) and collected by centrifugation at 1000× *g* rpm for 5 min, and resuspended in a complete medium. The cells were counted using an Auto Cellometer Mini (Nexcelom Bioscience, Lawrence, MA, USA). The cells were seeded into 96-well plates at a density of 1 × 104 cells per well. The relative cell activity was determined by the cell counting kit 8 assay (Beijing Lan Y Science & Technology, Beijing, China). The plates were incubated at 37 °C for 10 min. The absorbance of every well was determined at a wavelength of 450 nm.

### 2.4. GR Activity and GSH Content

The cells were collected using cell scratchers and broken by ultrasonic methods with PBS. After the mixture was centrifuged at 12,000× *g* for 10 min at 4 °C, the supernatant was collected for the following assay. The products of GR activity (A062-1) and GSH content (A006-2) were purchased from Nanjing Jiancheng Technology Co. Ltd. (Nanjing, China). Detection was performed according to the kit instructions.

### 2.5. Cell Apoptosis

Cell apoptosis was determined by the Annexin V/PI apoptosis detection kit (Solarbio). Annexin V can bind to phosphatidylserine on the surfaces of HepG2 apoptotic cells and is detected with a flow cytometer in the FITC channel, as a marker for apoptosis. PI can enter the dying or dead cells and is detected with a flow cytometer in PE channel, as a marker for cell death. The cells were segregated into four quadrants: Annexin V−/PI− (viable cells), Annexin V+/PI− (early apoptotic cells), Annexin V−/PI+ (necrotic cells), and Annexin V+/PI+ (late apoptotic cells). HepG2 cells were digested with trypsinization (Procell), centrifuged at 1000× *g* rpm for 5 min, and then collected. Subsequently, the cells were washed and resuspended in pre-cooled PBS and centrifuged for recollection. This elution step was repeated twice. Then, the cells were resuspended in 1 × binding buffer and incubated with Annexin V and PI staining solution at room temperature for 5 min. Cellular apoptosis was assessed via flow cytometry (Mindray, Shenzhen, China). Analysis was performed using Flow Jo software version 9.3 (Tree Star, Ashland, OR, USA) [38,39,40].

### 2.6. RNA Analysis and Real-Time PCR

The RNA extraction obtained from HepG2 cells was accomplished using 1 mL of Trizol (Beijing Lan Y Science & Technology) per well in 6-well plates. Reverse transcription was performed using the PrimeScript^TM^RT Master Mix kit (TaKaRa, Otsu, Japan); real-time PCR was conducted using the TB Green^®^ Premix Ex Taq^TM^ Ⅱ kit (TaKaRa, Otsu, Japan) and tested with the ABI 7500 Real-Time PCR System. The mRNA expressions of the target genes included *CHOP*, *78 kDa glucose-regulated protein* (*GRP78*), *activating transcription factor 6* (*ATF6*), *protein disulfide isomerase family A member 6* (*PDIA6*), *BAX*, *BCL2*, and *CASP3*. Primers were designed with Primer Premier 6 (Premier Biosoft, CA, USA) and the primer sequences are illustrated in Table 2. The results were normalized to the *β-actin* gene expression, and the mRNA expression was calibrated with the CON value. Fluorescence results were calculated in relation to the *β-actin* CT value using the 2ΔΔCT method [41,42,43].

### 2.7. Western Blot Analysis

Western Blot analysis was performed, as previously described [44]. The cell proteins were extracted using a RIPA extraction buffer (Solarbio); approximately 30 mg of protein samples were separated on 4~20% Bis-Tris SurePAGETM gels (Genscript, Nanjing, China) and electro-transferred to PVDF membranes (Pall, Pensacola, FL, USA). Subsequently, the membranes were blocked with 5% nonfat skimmed milk (BD Difco, Sparks, MD, USA) for 1 h at room temperature and incubated overnight at 4 °C with primary antibodies against CHOP (1:1000, ABclonal, Wuhan, China), 78 kDa glucose-regulated protein (GRP78, 1:1000, CST, Framingham, MA, USA), activating transcription factor 6 (ATF6, 1:1000, CST), protein disulfide isomerase family A member 6 (PDIA6, 1:1000, ABclonal), BAX (1:1000, ABclonal), BCL2 (1:1000, ABclonal), CASP3 (1:1000, CST), and β-tubulin (1:5000, Huaxingbio, Beijing, China). After washing with 1 × TBST three times, the membranes were incubated at room temperature for 1 h with appropriate secondary antibodies (1:5000 dilution, goat anti-rabbit, Solarbio; 1:5000 dilution, goat anti-mouse, Solarbio). Subsequently, the blot bands were visualized with an ECL reagent (Beijing Lan Y Science & Technology). Then, the optical densities of the blot bands were analyzed using Tanon Gis 1D software (Tanon Sciences & Technology, Beijing, China). Finally, the protein expression was normalized by β-tubulin and calibrated with the CON value.

### 2.8. Data and Statistical Analyses

The data were analyzed using the two-tailed Student’s *t*-test or one-way ANOVA using SAS 9.4 (SAS Institute Inc, Cary, NC, USA). All the data were expressed as means ± standard deviation (SD) and presented using GraphPad Prism Version 9.3 (GraphPad, La Jolla, CA, USA). *p* < 0.05 was considered statistically significant.

## 3. Results

### 3.1. RD Inhibited the Cell Proliferation of HepG2cells and Decreased Relative Cell Viability

The cell number in the RD group was significantly lower than that in the CON group at 2, 3, 4, and 5 d (Figure 1A), indicating that RD inhibited cell proliferation of HepG2 cells. Moreover, a negative growth appeared in the RD group for 5 d (Figure 1A). Compared to the CON group, RD significantly reduced the relative cell viability with time and gradually decreased from 94.5% on day 1 to 38.5% on day 5 (Figure 1B).

### 3.2. RD Inhibited GR Activity and Decreased GSH Content

In this experiment, the cellular GR activity and GSH content were analyzed to explore the effects of RD on the redox equilibrium in the ER of HepG2 cells. The results show that the GR activity and GSH content in the RD group are approximately three times lower than those in the CON group (Figure 2A,B), indicating that RD disrupts ER redox homeostasis.

### 3.3. RD-Induced ER Stress and Apoptosis

The TEM results show that RD significantly increases the swelling of ER lumens (Figure 3A). In comparison to the CON group, the diameter of the ER lumen expanded 1.88-fold in the RD group (Figure 3B), while its area enlarged 4.93-fold (Figure 3C). Additionally, ruptured mitochondria cristae in RD cells were also observed. The cell apoptosis rate significantly increased on the third day of riboflavin depletion than that of the riboflavin-sufficient group (Figure 4A,B). Moreover, the total cell apoptosis rate (early apoptosis rate + late apoptosis rate) increased from 3.44% to 26.2% after riboflavin depletion from days two to five (Figure 4A,B).

HepG2 cells were cultured in the RD medium for 4 d, where mediators of ER stress-induced apoptosis were detected, such as CHOP, GRP78, ATF6, CASP3, and BAX. The above-described mRNA expressions were significantly upregulated, and the expression of the antiapoptotic gene BCL2 was significantly downregulated in the RD group compared to the CON group (Figure 5A). The protein expression of CHOP, GRP78, and ATF6 was 2~3 fold, and CASP3 presented a 1.32-fold-higher expression in the RD group than in the CON group (Figure 5B). The mRNA and protein expressions of PDIA6 decreased in riboflavin-depleted cells compared to riboflavin-sufficient cells (Figure 5A,B). Taken together, these results indicate that RD can affect protein folding, trigger ER stress, and eventually induce cellular apoptosis.

### 3.4. 4-PBA Attenuated ER Stress and Apoptosis Induced by RD

To investigate whether RD-induced apoptosis was mediated by ER stress, 4-PBA was utilized to alleviate ER stress. The results show that the administration of -PBA significantly increases cell numbers and relative cell viability both in the RD and CON groups. Following treatment with 5 mM 4-PBA, the cell number slightly increased by 24% and the relative cell viability slightly increased by 14% in the CON group, while those values increased considerably, by 189% and 68%, in the RD group, respectively (Figure 6A,B). The cell number of the RD group following treatment with 2.5 mM 4-PBA was already equal to the CON group (Figure 6A). Moreover, when riboflavin-depleted cells were treated with 5 mM 4-PBA, the relative cell viability slightly improved, but remained significantly lower than that in the CON group (93.7% vs. 100%) (Figure 6B).

As shown in (Figure 6C,D), 4-PBA administration did not affect the cell apoptosis rate in the riboflavin-sufficient cells. However, the administration of 5 mM 4-PBA could alleviate apoptosis induced by RD, and there was no difference in the early and late apoptosis rates between the RD and CON groups (Figure 6C,D). Moreover, the results show that 4-PBA treatment decreases the mRNA and protein expressions of ER stress markers (ATF6, GRP78, and CHOP) and proapoptotic markers (BAX and CASP3), while upregulating the antiapoptotic gene and protein expressions (BCL2) in the RD group (Figure 7A,C) compared to the untreated group. In contrast, 4-PBA administration did not affect the apoptosis-related gene and protein expressions in the CON group, which further validates the notion that 4-PBA treatment did not affect apoptosis in the CON cells. These results indicate that 4-PBA attenuates RD-induced apoptosis in HepG2 cells by inhibiting ER stress, implying that RD promotes apoptosis is mainly mediated by ER stress.

### 3.5. Knockdown of CHOP Attenuated RD-Induced ER Stress and Apoptosis

Since CHOP is a key stress-associated protein regulating ER stress-mediated apoptosis [45,46,47], we examined whether RD-induced ER stress-mediated apoptosis was regulated by CHOP more extensively. Following the CHOP knockdown, the cell number markedly increased in the RD group. In contrast, CHOP siRNA did not affect the cell number of the riboflavin-sufficient group. Similarly, compared to the RD-blank control (BC) or RD-NC group, the relative cell viability in the RD-si group also markedly increased, but was still significantly lower than that of the CON-BC and CON-NC groups (Figure 8A,B). As presented in Figure 8B, the apoptosis rate of riboflavin-sufficient cells slightly decreased from 6.0% to 4.2% after CHOP siRNA transfection. However, the total cell apoptosis rate of riboflavin-deficient cells following siRNA transfection dramatically decreased by 58.1% compared to the RD-BC group, which was not different for the CON-BC and CON-NC groups (Figure 8C,D). Consistent with the apoptosis-rate results, the gene and protein expressions of CASP3 and BAX decreased, and the BCL2 gene and protein expression increased in the RD-transfected group, compared to the RD-BC or RD-NC groups (Figure 9A,C). However, these genes and proteins presented no significant change in CON cells. These data suggest that RD-induced ER stress-mediated apoptosis via the CHOP pathway.

## 4. Discussion

Riboflavin is involved in energy metabolism [48], fat metabolism [49,50], and antioxidative effects [5,51]. As a precursor of FMN and FAD, riboflavin has been reported to participate in protein folding through Ero1 and GR [7]. A deficiency in riboflavin decreased the expressions of Ero1 and PDI [19,29] and reduced GR activity and GSH content [3,19,27], which could hamper the occurrence of correct protein-folding activity in the ER. It has been reported that IL-2 and ApoB-100 appeared as unfolded/misfolded in the case of RD [15,17,29]. Unfolded/misfolded proteins induce ER stress through an unfolded-protein response [52]. The depletion of riboflavin caused ER stress [15,17,18] and increased the ER stress-maker (CHOP) expression [16,17,18]. It has been well established that persistent and prolonged ER stress can trigger apoptosis [32,53], and the transcription factor CHOP is a major regulator of ER stress-induced apoptosis [54,55]. Indeed, the depletion of riboflavin-induced cell apoptosis in both chicken embryos [19] and cells [15,16,17,18]. Therefore, we hypothesized that RD-induced apoptosis is mediated by ER stress and the CHOP pathway. In the present study, we performed experiments to test this hypothesis.

HepG2 cells have a high demand for riboflavin, and an RD model was previously successfully established in this cell line [16]. Thus, HepG2 cells were used for constructing an experimental RD model to identify the effects of RD on ER stress and apoptosis. In line with the results obtained from a previous study, RD inhibited cell proliferation and cell viability [19], as well as decreased flavoprotein GR activity and GSH content [3,19,27]. Apoptosis is one of the main factors that affects cell proliferation. In that regard, the appearance of dilated ER indicated direct morphological evidence of RD-induced ER stress, and an increased rate of apoptosis was detected in the RD group. ATF6 is one of the ER stress-sensor proteins, regulated by GRP78, and activates the downstream gene CHOP to trigger apoptosis [32,56]. In the present study, RD increased the expression levels of a series of ER stress factors (ATF6, GRP78, and CHOP) and proapoptotic factors (BAX and CASP3). It decreased the expression of the antiapoptotic factor (BCL2), while the levels of these markers were attenuated by riboflavin-sufficient cells. These results are consistent with the typical features of ER stress and apoptosis. Thus, these results are in line with those obtained from previous studies, which further demonstrates that RD can induce ER stress and apoptosis [15,16,17,18,19].

A cell model of the inhibition of ER stress was constructed by an ER inhibitor to explore whether RD-induced apoptosis was mediated by ER stress. 4-PBA, a molecular chemical chaperone, attenuates ER stress in multiple cell types [57,58]. Theoretically, if ER stress is responsible for RD-induced apoptosis, then cell apoptosis could be suppressed in the RD group after blocking ER stress. As expected, the addition of 4-PBA inhibited RD-induced ER stress, indicated by the decreased expressions of ATF6, GRP78, and CHOP, which is in line with the results obtained from previous studies [59,60,61]. In the riboflavin-depleted cells, 4-PBA treatment decreased the rate of apoptosis, decreased the expressions of BAX and CASP3, and increased the BCL2 expression, which could block RD-induced apoptosis, while the state of apoptosis was stable without treating 4-PBA. We also observed that there was no significant effect on apoptosis treated by 4-PBA, and no ER stress occurred in riboflavin-sufficient cells. Our data show that 4-PBA abolishes unfolded/misfolded protein-induced ER stress and apoptosis, and thus substantially increases relative cell viability and promotes cell proliferation in riboflavin-deficient cells. Moreover, riboflavin-sufficient cells treated with 4-PBA could slightly promote protein folding, cell viability, and cell proliferation. These results are in accordance with the feature of ER stress-induced apoptosis [60,61]. These results determine that the inhibition of ER stress may prevent RD-induced apoptosis by suppressing proapoptotic pathways, which could demonstrate RD-induced apoptosis mainly mediated by ER stress.

CHOP is an ER-stress hallmark and an inducer of ER stress-induced apoptosis [45,46,47]. A previous study revealed that *CHOP* overexpression could promote apoptosis [62], while *CHOP* knockout decreased cell apoptosis [63]. There is some evidence that CHOP promotes apoptosis by inhibiting BCL2 and activating BAX and CASP3 [33,35]. The present study demonstrated that the 4 d depletion of riboflavin-induced apoptosis was mediated by ER stress. Theoretically, if the CHOP pathway was responsible for RD-induced apoptosis, then cell apoptosis could be suppressed in the RD group following CHOP knockdown. We thus used *siRNA* to knockdown *CHOP* to probe whether RD-induced apoptosis was regulated by CHOP. The gene and protein expressions of CHOP decreased, as expected from the *siRNA* transfection. For the RD-group cells, our results show that *CHOP* knockdown decreases the rate of apoptosis, inhibits BAX and CASP3, and activates BCL2, which could block RD-induced apoptosis, while maintaining the apoptosis state without transfecting *CHOP siRNA*. The relative cell viability considerably increased in RD cells by *CHOP siRNA*-inhibited apoptosis, and recovered cell numbers without presenting a difference in the CON cells. The results show a significantly reduced early apoptosis rate and a significantly increased relative cell viability in the CON cells after being transfected with *CHOP siRNA*, but the magnitude of this effect is much smaller than in the RD cells. These results are in accordance with the feature of CHOP-mediated apoptosis [63]. These results determine that the inhibition of *CHOP* may prevent RD-induced apoptosis by blocking the activation of proapoptotic pathways, demonstrating that RD induces apoptosis in a CHOP-dependent manner.

Additionally, these interesting results determine that, regardless of ER stress inhibition or CHOP knockdown, the relative cell viability in the riboflavin-depleted cells is consistently less than that in the riboflavin-sufficient cells. We analyzed three possible explanations for this result:Even though apoptosis was inhibited by CHOP interference, unfolded/misfolded proteins were still retained in the ER, and ER stress persisted in the riboflavin-depleted cells.Riboflavin depletion resulted in the low expression of flavoproteins, leading to metabolic disorders.Mitochondrial damage of RD-group cells was observed in TEM (Figure 3A), which may inhibit cell viability. Indeed, previous studies have determined that cell viability could be influenced by ER stress [64,65] and mitochondrial damage [66,67].

## 5. Conclusions

In summary, in the present study, we investigated how riboflavin deficiency for 4 d induced apoptosis in HepG2 cells. The results demonstrate that RD induces apoptosis through the ER stress pathway. Specifically, RD hinders the correct folding of proteins to trigger ER stress, resulting in the activation of the CHOP pathway and subsequent apoptosis.

## Figures and Tables

**Figure 1 nutrients-14-03356-f001:**
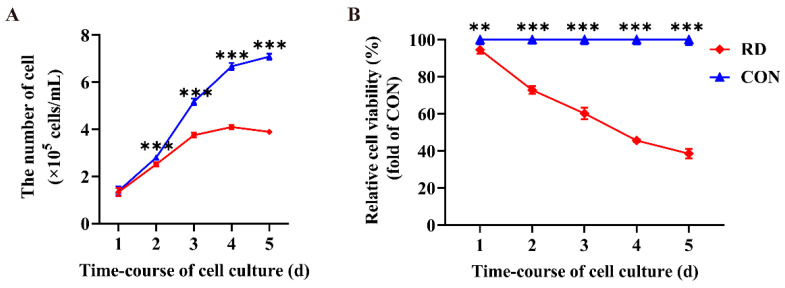
Riboflavin deficiency (RD) decreases cell numbers and relative cell viability. (**A**) Cell numbers in the RD and control (CON) groups vary over time. (**B**) Relative cell viability in the RD and CON groups vary over time. Values are expressed as means ± SD. ** indicates that the differences are significant at *p* < 0.01; *** indicates that the differences are significant at *p* < 0.001. Each treatment was performed in six replicates (*n* = 6).

**Figure 2 nutrients-14-03356-f002:**
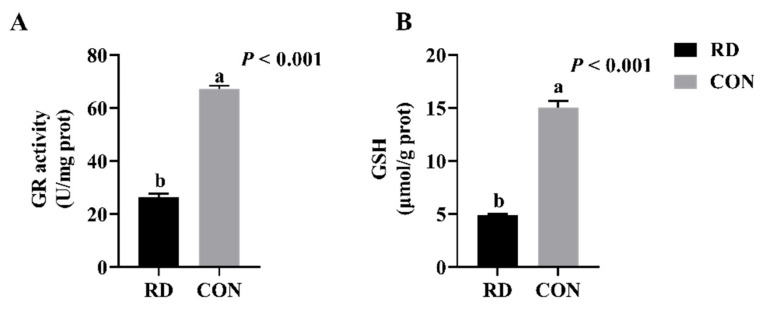
RD decreases glutathione reductase (GR) activity and glutathione (GSH) content. (**A**) GR activity of HepG2 cells cultured at 4 d. (**B**) GSH content of HepG2 cells cultured at 4 d. Values are means ± SD. Different letters indicate significant differences (*p* < 0.05). Each treatment is performed in six replicates (*n* = 6).

**Figure 3 nutrients-14-03356-f003:**
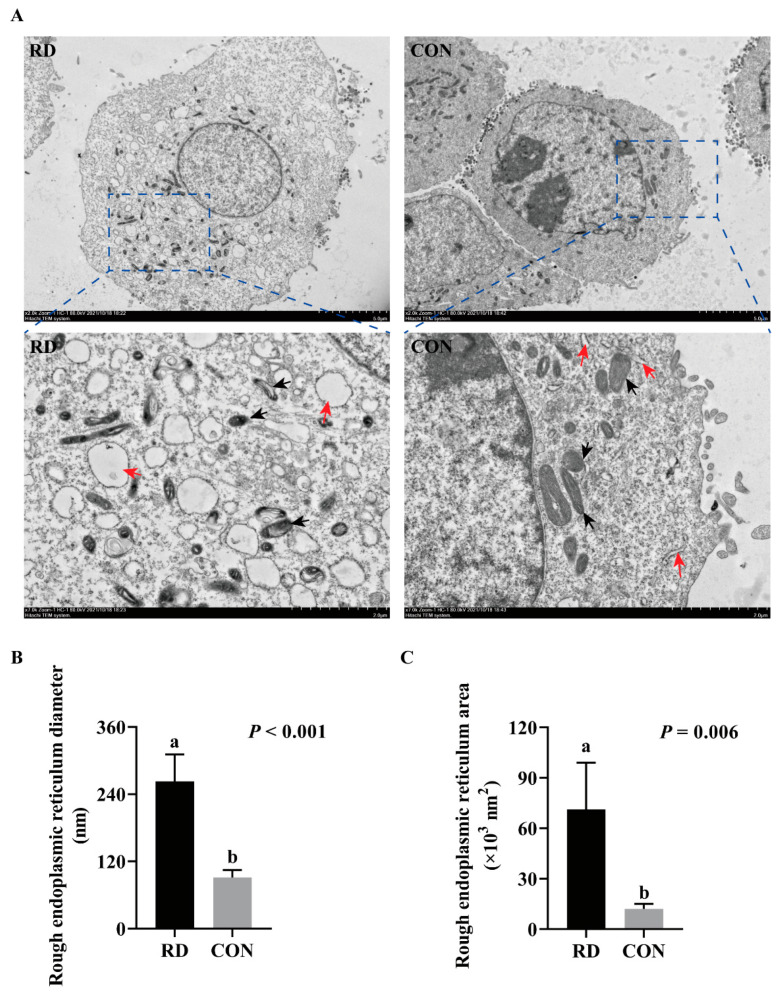
RD induces the swelling of the endoplasmic reticulum (ER). (**A**) The cellular ultramicroscopic structures observed by transmission electron microscopy (TEM); black arrowheads point to ER, and red arrowheads point to mitochondria. Effect of RD on the diameter (**B**) and area (**C**) of rough ER. Values are expressed as means ± SD. Different letters indicate significant differences (*p* < 0.05). Each treatment is performed in four replicates (*n* = 4).

**Figure 4 nutrients-14-03356-f004:**
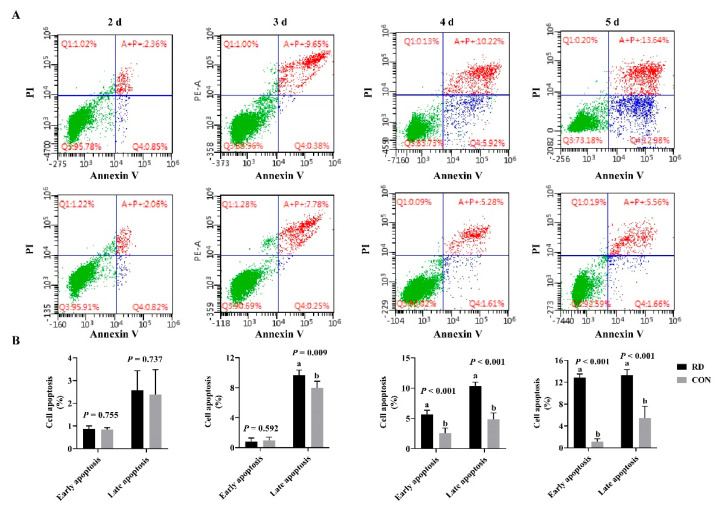
RD-induced cell apoptosis. (**A**) Cellular apoptosis determined by flow cytometry. (**B**) Results of cell apoptosis analysis. Different letters indicate significant differences (*p* < 0.05). Each treatment is performed in six replicates (*n* = 6).

**Figure 5 nutrients-14-03356-f005:**
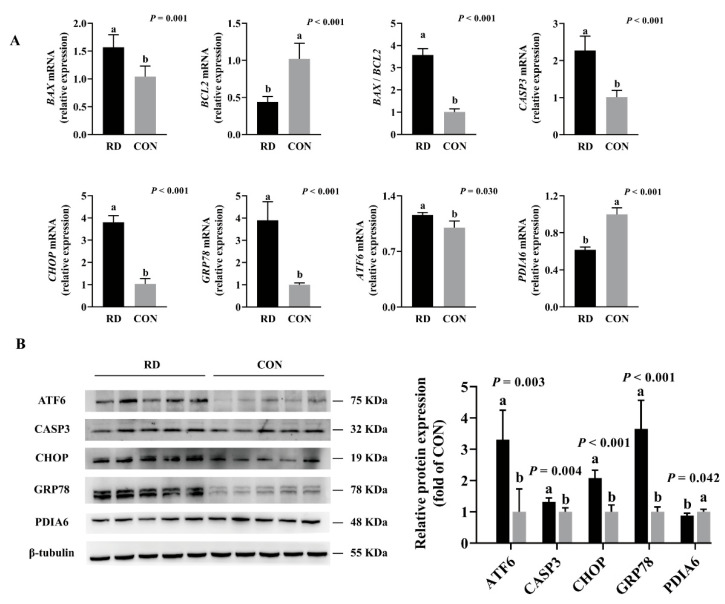
Real-time PCR and Western blot analyses verify gene and protein expressions of HepG2 cells. (**A**) Results of RD on the mRNA expression of ER stress and apoptosis. Representative real-time PCR of *BAX*, *BCL2*, *BAX* /*BCL2*, *CASP3*, *CHOP*, *GRP78*, *ATF6*, and *PDIA6* are displayed. Each treatment is performed in six replicates (*n* = 6). (**B**) Results of RD on the protein expression of ER stress and apoptosis. Representative Western blot of ATF6, CASP3, CHOP, GRP78, and PDIA6 are depicted. Different letters indicate significant differences (*p* < 0.05). Each treatment is performed in five replicates (*n* = 5).

**Figure 6 nutrients-14-03356-f006:**
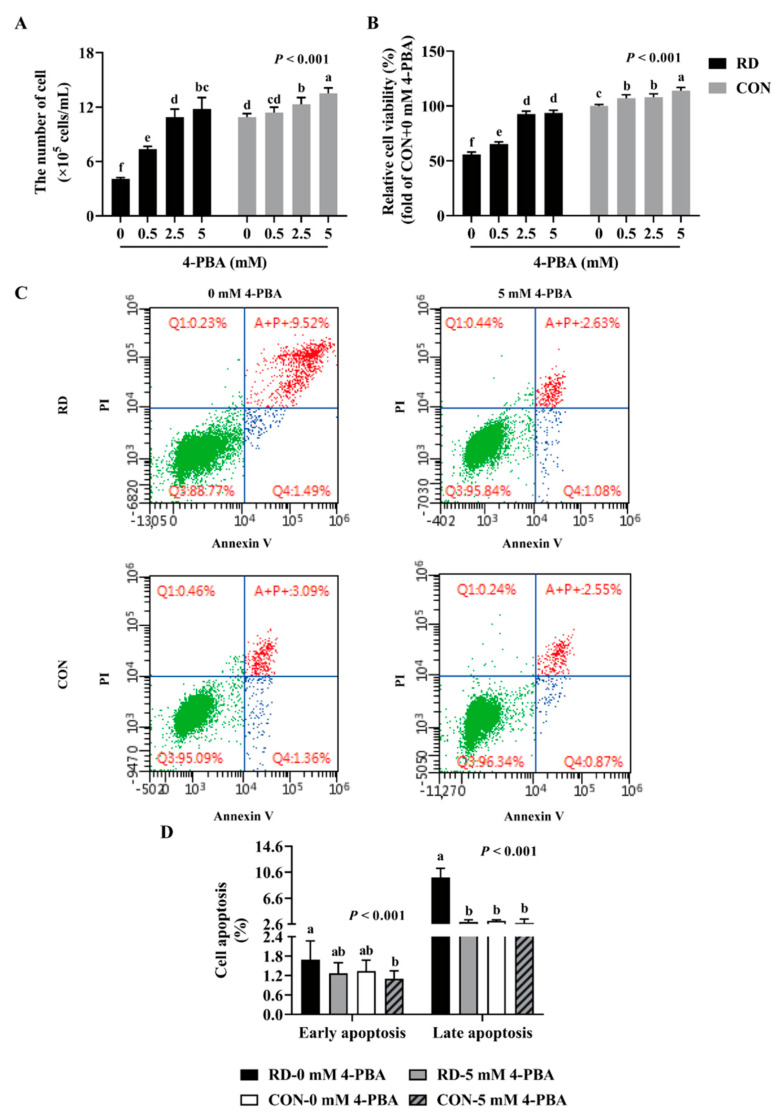
4-phenyl butyric acid (4-PBA) can prevent RD-induced cell apoptosis. Effect of different concentrations (0, 0.5, 2.5, and 5 mM) of 4-PBA on cell number (**A**) and relative cell viability (**B**) in riboflavin-deficient cells. (**C**) Effect of RD on cell apoptosis following treatment with 4-PBA (5 mM) in HepG2 cells. (**D**) Results of cell apoptosis analysis. Different letters indicate significant differences (*p* < 0.05). Each treatment is performed in six replicates (*n* = 6).

**Figure 7 nutrients-14-03356-f007:**
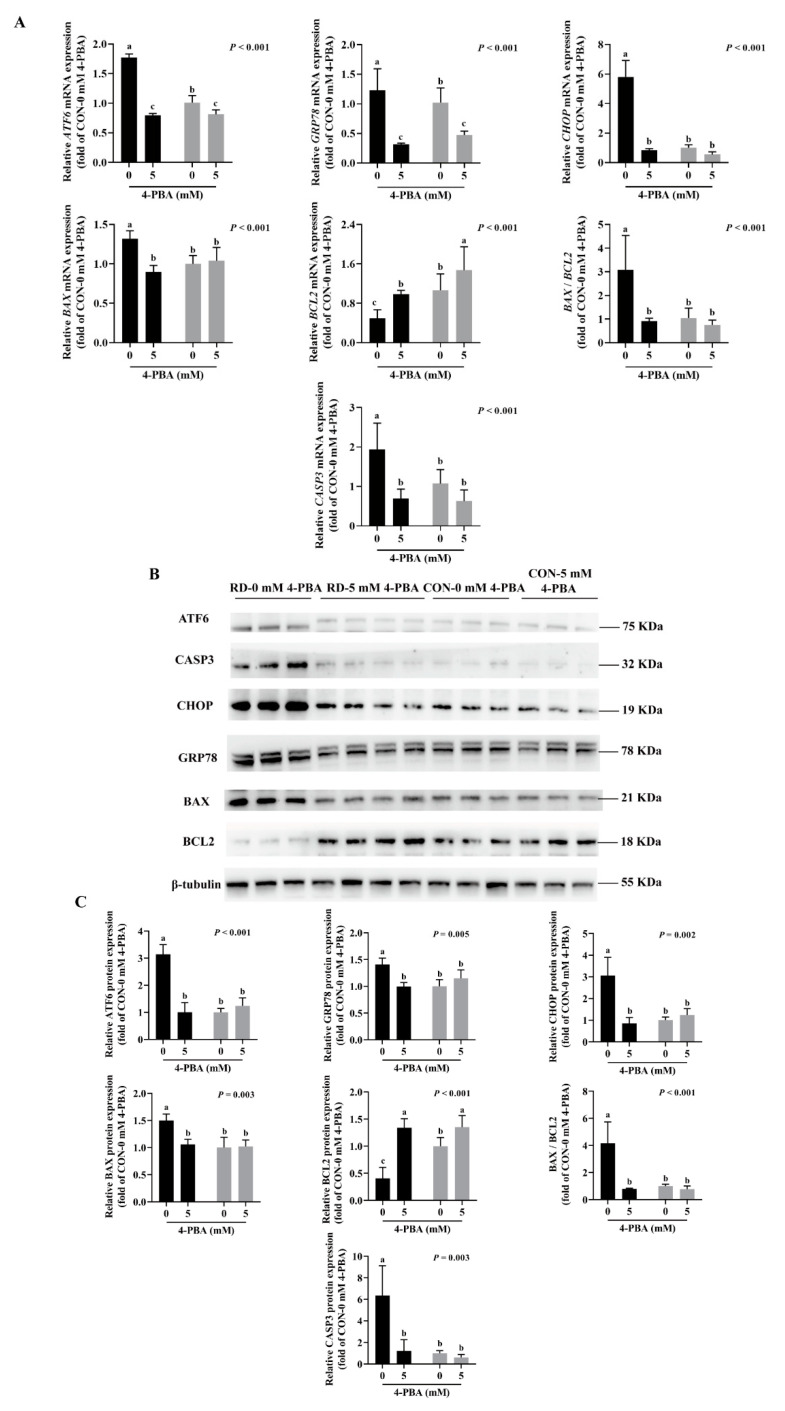
Expression analysis of ER stress- and apoptosis-related genes and proteins in riboflavin-deficient cells treated with 4-PBA. (**A**) Results of 4-PBA treatment on the mRNA expression of riboflavin-deficient cells. Representative real-time PCR of *ATF6*, *GRP78*, *CHOP*, *BAX*, *BCL2*, *BAX* /*BCL2*, and *CASP3* are depicted. Each treatment is performed in six replicates (*n* = 6). (**B**) Effect of RD on protein expression following treatment with 4-PBA (5 mM) in HepG2 cells. (**C**) Western blot analysis is performed. Representative Western blots of ATF6, GRP78, CHOP, BAX, BCL2, BAX /BCL2, and CASP3 are illustrated. Different letters indicate significant differences (*p* < 0.05). Each treatment is performed in three to four replicates (*n* = 3~4).

**Figure 8 nutrients-14-03356-f008:**
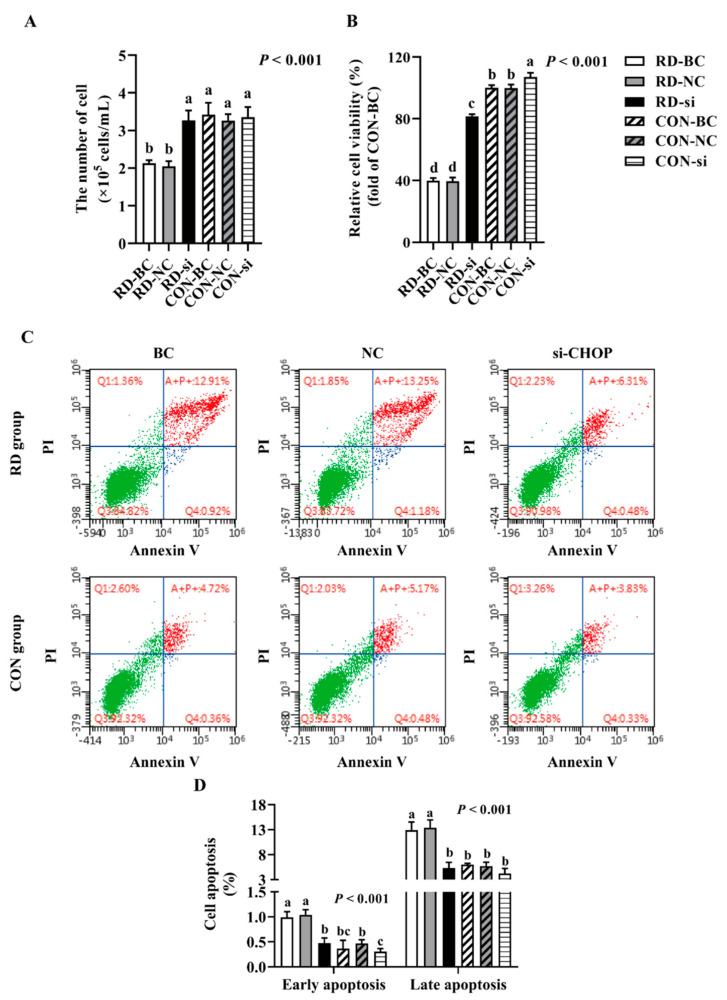
*CHOP* interference can resist RD-induced cell apoptosis. Effect of *CHOP* interference on cell number (**A**) and relative cell viability (**B**) in riboflavin-deficient cells. Each treatment is performed in six replicates (*n* = 6). (**C**) Effect of RD on cell apoptosis following treatment with *CHOP siRNA* in HepG2 cells. (**D**) Results of cell apoptosis analysis. Different letters indicate significant differences (*p* < 0.05). Each treatment is performed in three to six replicates (*n* = 3~6).

**Figure 9 nutrients-14-03356-f009:**
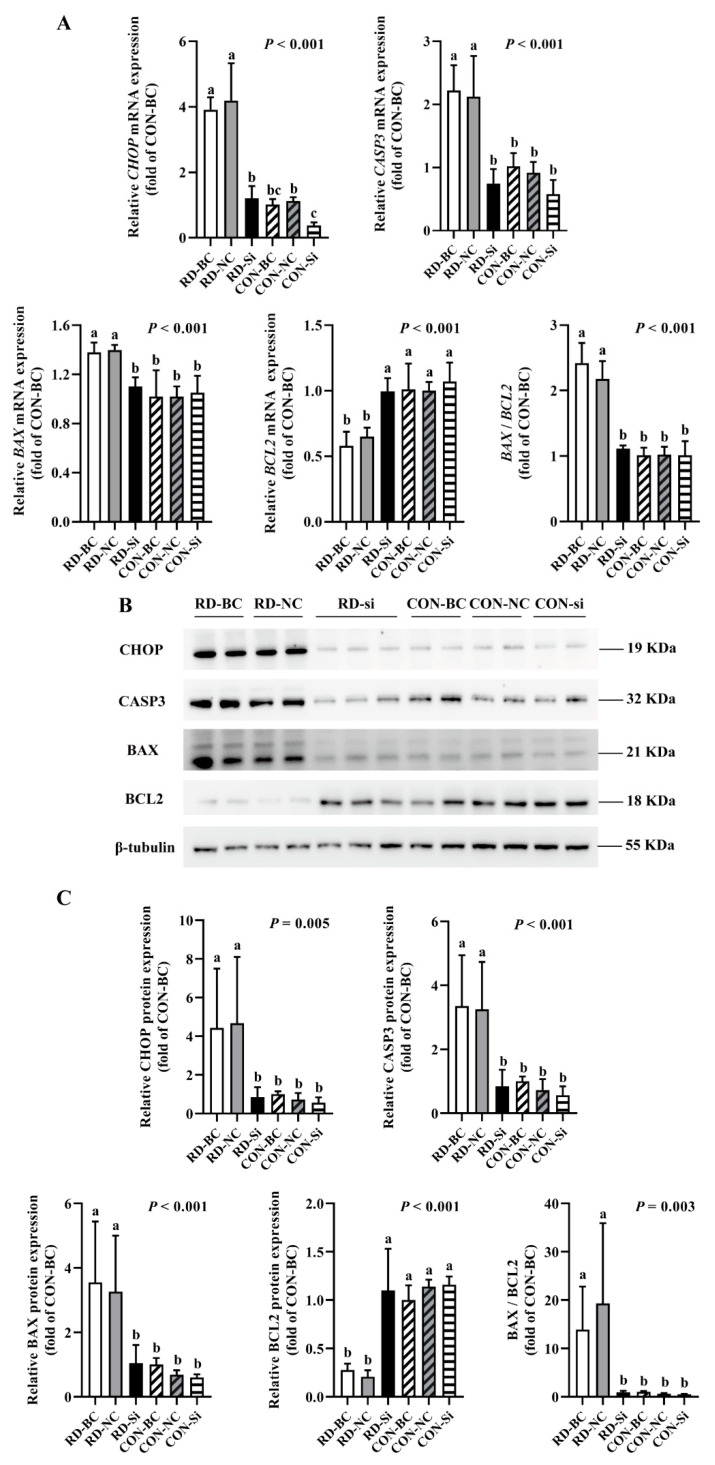
Gene and protein expression analysis of apoptosis-related effects in riboflavin-deficient cells treated with *CHOP siRNA*. (**A**) Results of *CHOP siRNA* on the mRNA expression of riboflavin-deficient cells. Representative real-time PCR of *CHOP*, *CASP3*, *BAX*, *BCL2*, and *BAX* /*BCL2* are presented. Each treatment is performed in five replicates (*n* = 5). (**B**) Results of *CHOP siRNA* on protein expression of riboflavin-deficient cells. (**C**) Western blot analysis is performed. Representative Western blot of CHOP, CASP3, BAX, BCL2, and BAX /BCL2 are presented. Different letters indicate significant differences (*p* < 0.05). Each treatment is performed in four or six replicates (*n* = 4 or 6).

**Table 1 nutrients-14-03356-t001:** siRNA primer sequence.

siRNA	Sense	Antisense
*CHOP*	5′-GAACAGGAGAAUGAAAGGAAATT-3′	5′-UUUCCUUUCAUUCUCCUGUUCTT-3′
*NC*	5′-UUCUCCGAACGUGUCACGUTT-3′	5′-ACGUGACACGUUCGGAGAATT-3′

Abbreviations: *NC*, negative control.

**Table 2 nutrients-14-03356-t002:** Primers used for the target and reference genes.

Genes	Forward	Reverse
*CHOP*	5′-ACCAAGGGAGAACCAGGAAACG-3′	5′-TCACCATTCGGTCAATCAGAGC-3′
*GRP78*	5′-CGGGCAAAGATGTCAGGAAAG-3′	5′-TTCTGGACGGGCTTCATAGTAGAC-3′
*ATF6*	5′-AGCGGAGCCACTGAAGGAAGATA-3′	5′-GCGTTGGTACTGTCTGAATAATGATGG-3′
*PDIA6*	5′-GGAGGTCAGTATGGTGTTCAGGGAT-3′	5′-CTGCCACCTTGGTAATCTTCTGGTC-3′
*BAX*	5′-GGCCCACCAGCTCTGAGCAGA-3′	5′-GCCACGTGGGCGGTCCCAAAGT-3′
*BCL2*	5′-GTGGAGGAGCTCTTCAGGGA-3′	5′-AGGCACCCAGGGTGAGCAA-3′
*CASP3*	5′-TCTGACTGGAAAGCCGAAACT-3′	5′-AGTGACTGGATGAACCATGAC-3′
*β-actin*	5′-AATCGTGCGTGACATTAAGGAGAAG-3′	5′-CAGGAAGGAAGGCTGGAAGAGTG-3′

Abbreviations: *CHOP*, C/EBP homologous protein; *GRP78*, 78 kDa glucose-regulated protein; *ATF6*, activating transcription factor 6; *PDIA6*, protein disulfide isomerase family A member 6; *BAX*, *BCL2*-associated X, apoptosis regulator; *BCL2*, B-cell lymphoma 2, apoptosis regulator; *CASP3*, Caspase 3.

## Data Availability

Not applicable.

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
