# Peer review of "Riboflavin (Vitamin B2) Deficiency Induces Apoptosis Mediated by Endoplasmic Reticulum Stress and the CHOP Pathway in HepG2 Cells"

_nutrients, 2022, doi:10.3390/nu14163356_

Round 1
Reviewer 1 Report
The results presented in the manuscript entitled “Riboflavin (vitamin B2) Deficiency Induces Apoptosis Mediated by Endoplasmic Reticulum Stress and the CHOP Pathway in HepG2 Cells” are in a logical sequence to that contain data to inform the readers. The manuscript is interesting to publication in “Nutrients” after revision.
1) At first, some grammatical points can be seen in the text of the manuscript.
2) The “Abstract section” should be categorized as background, methods, results, and discussion.
3) The novelty and hypothesis of the research must be mentioned at the end of the “Introduction section”.
4) All primer sequences must be presented as table.
5) The manuscript is not well referenced. All parts of the “Methods section” should be explained in more detail. For example, western blotting as well as Real-time PCR methods were elaborately explained in the research by Adibkia et al. (2021). You can use and refer the following paper which explain elaborately and completely these methods:
ü Silver nanoparticles induce the cardiomyogenic differentiation of bone marrow derived mesenchymal stem cells via telomere length extension. Beilstein journal of nanotechnology. 2021 Aug 2;12(1):786-97.
6) Also, cell apoptosis assessment was elaborately explained in the research by Fathi et al. (2021). You can use and refer the following paper which explain elaborately and completely this method:
ü Mesenchymal stem cells promote caspase-3 expression of SHSY5Y neuroblastoma cells via reducing telomerase activity and telomere length. Iranian Journal of Basic Medical Sciences. 2021 Nov 1; 24(11).
7) Other mentioned methods should be referenced by the appropriate references.
8) Since the discussion section is one of the most important parts of the paper, this section must be improved with more attention and explanation. In the discussion section, results must be compared with another results from previous studies.
Reviewer 2 Report
The nutrients-1862583 is well-done paper to proved Vit B2 deficiency could cause hepatic ER stress. Overall data are sufficient to be published to the Nutrients; however, there is a minor comment as a reviewer.
- Please provide the Vit B2 absorption data by HepG2 cells. If cellular absorption data is not provided, then anti-ER stress effects may possibly due to the Vit B2 in the culture media rather than intracellular.
